# Predicting the taxonomic and environmental sources of integron gene cassettes using structural and sequence homology of *attC* sites

Timothy M. Ghaly [1✉], Sasha G. Tetu [2,3] & Michael R. Gillings [1,3]

Integrons are bacterial genetic elements that can capture mobile gene cassettes. They are mostly known for their role in the spread of antibiotic resistance cassettes, contributing significantly to the global resistance crisis. These resistance cassettes likely originated from sedentary chromosomal integrons, having subsequently been acquired and disseminated by mobilised integrons. However, their taxonomic and environmental origins are unknown. Here, we use cassette recombination sites (*attC*s) to predict the origins of those resistance cassettes now spread by mobile integrons. We modelled the structure and sequence homology of 1,978 chromosomal *attC*s from 11 different taxa. Using these models, we show that at least 27% of resistance cassettes have *attC*s that are structurally conserved among one of three taxa (Xanthomonadales, Spirochaetes and Vibrionales). Indeed, we found some resistance cassettes still residing in sedentary chromosomal integrons of the predicted taxa. Further, we show that *attC*s cluster according to host environment rather than host phylogeny, allowing us to assign their likely environmental sources. For example, the majority of β-lactamases and aminoglycoside acetyltransferases, the two most prevalent resistance cassettes, appear to have originated from marine environments. Together, our data represent the first evidence of the taxonomic and environmental origins of resistance cassettes spread by mobile integrons.

[1] Department of Biological Sciences, Macquarie University, Sydney, Australia. [2] Department of Molecular Sciences, Macquarie University, Sydney, Australia. [3] ARC Centre of Excellence in Synthetic Biology, Macquarie University, Sydney, Australia. ✉email: timothy.ghaly@mq.edu.au

ntegrons are bacterial genetic elements that can insert and excise mobile gene cassettes by site-specific recombination. They are mostly known for the accumulation and spread of antibiotic resistance gene cassettes among Gram-negative bacteria[1], although it is clear that they can mobilise and rearrange a diverse range of cassettes that encode functions well beyond clinical relevance[2]. The recruitment of gene cassettes by an integron is mediated by its core functional gene, the integron integrase (IntI). Once inserted, gene cassettes form part of a cassette array that can vary considerably in size, ranging from one or two cassettes to more than three hundred[3–5]. Cassette arrays are highly variable regions, in which the content and arrangement of genes can be changed by IntI activity, often induced by environmental stress[6,7]. Consequently, integrons provide genomic plasticity and adaptation on demand[8].

Integrons have been classified into two main groups. The first are mobile integrons, which comprise five known IntI classes that have become embedded into mobile elements. These integrons, particularly the class 1 integron, are of clinical significance, often harbouring multiple antibiotic-resistance cassettes. The class 1 integron is believed to have become mobilised from a Betaproteobacterial chromosome in the early 20th century[1,9]. Since that time, derivatives of this ancestral element have spread into more than 70 clinically important bacteria and have acquired more than 130 different resistance genes that confer resistance to most classes of antibiotics[10,11]. Mobile class 1 integrons have been found on every continent, including Antarctica, and are ubiquitous in the microbiomes of humans and agricultural animals[11,12]. This successful colonisation means that up to $10^{23}$ copies of class 1 integrons are shed into the environment every day via human and agricultural waste[12].

Class 1 integrons, together with the other four mobile classes, vector almost all antibiotic resistance gene cassettes that can be detected in bacterial genomes[5]. Consequently, their success has been largely driven by strong antimicrobial selection. They differ considerably from sedentary chromosomal integrons (SCIs), which represent the ancestral state of integrons[13]. These largely carry gene cassettes of unknown functions[2], and are present in ~17% of sequenced genomes[14].

Gene cassettes are mobilisable elements that carry a cassette-associated recombination site (attC). The attC site allows insertion into a cassette array at the integron-associated recombination site (attI) (Fig. 1A). Insertion involves the recombination between attI and only the bottom strand of the attC site[15,16], forming an atypical Holliday junction (Fig. 1B). As only one strand of the attC is involved in recombination, the Holliday junction cannot be resolved by the typical second-strand exchange. Instead, it is resolved by replication of the entire recombinant molecule[17] (Fig. 1B). Folding of the attC single strand is permitted by the pairing of two sets of inverted repeats (R′ to R″ and L′ to L″) (Fig. 1C). For most attCs, however, the number of nucleotides involved in pairing extends beyond the R and L boxes, which influences the final attC secondary structure. As a consequence, the overall structure and length of attCs can vary considerably between different gene cassettes (Fig. 1D).

The efficiency of recombination is largely dependent on the folded hairpin structure of the bottom attC strand[18–20]. Further, different integron integrases can recognise different ranges of attC structures. In particular, the class 1 integron integrase (IntI1) has a broad attC specificity range[21,22]. This is likely to be one of the reasons for its remarkable success after it became mobilised, because it can gain access to, and recognise, diverse gene cassettes from broad phylogenetic sources. Indeed, cassettes associated with mobile class 1 integrons appear to have been acquired from diverse chromosomal contexts. This is evident from the inconsistent codon usage in their cassette open reading frames (ORFs)

and the considerable sequence and structural diversity of their attC sites[14]. In contrast, SCIs generally contain cassettes with highly similar attCs, and these share homology with other attCs in the same bacterial taxon[23–25]. This conservation provides an opportunity to predict the taxonomic origins of clinically important gene cassettes disseminated by mobile integrons. Hereafter, we use the term 'origin' to represent the SCI source from which the cassettes were acquired by mobile integrons.

Knowledge of the taxonomic origins of resistance cassettes, and what ecological and physiological traits are shared by these taxa, might allow us to predict environmental hotspots or conditions that contribute to the emergence of novel resistance cassettes[26]. This could suggest efficient mitigation strategies to prevent the spread of novel resistance genes into clinical settings.

Here, we modelled the conserved structure and sequence homology of attCs from distinct bacterial taxa. We used these taxon-specific models to predict the origins of resistance gene cassettes found on mobile integrons and provide evidence for some of these cassettes still residing in the SCIs of those taxa. Further, we show that attCs are more similar among bacteria that inhabit a similar environment, allowing us to predict the environmental sources of each resistance cassette. In particular, we show that the majority of β-lactamases and aminoglycoside acetyltransferases, the two most prevalent resistant cassettes, appear to have originated in marine bacteria. We also find that both the structure of attCs and the phylogeny of IntIs cluster according to the host environment rather than host phylogeny. We propose that this shared clustering pattern is the result of convergent and co-evolutionary processes. IntI-attC co-evolution would allow recombination specificity to be maintained within a taxon, while convergent evolution facilitates the successful exchange of cassettes between divergent taxa inhabiting the same environment.

## Results and discussion

**Efficacy of attC taxonomy predictions.** We used the sequence and structural homology of attC recombination sites, conserved within individual taxa, to predict the sources of gene cassettes spread by mobile integrons. To do this, we generated covariance models (CMs) built from sets of taxon-specific attCs obtained from the chromosomes of diverse bacteria. CMs are similar to profile hidden Markov models (HMMs) in that they both capture position-specific information about how conserved each column of an alignment is, and which nucleotides/residues are likely to occur. However, in a profile HMM, each position of the profile is treated independently, while in a CM, base-paired positions (when folded) are dependent on one another, therefore, modelling the covariation at these positions. This is necessary to assess correct base-pairing in secondary structure formation. Thus, both conserved sequence and secondary structure of taxon-specific attCs can be modelled.

We determined the specificity and sensitivity of each model in assigning the correct taxonomy to our complete attC dataset ($n = 2,352$). The efficacy of each CM was determined by its sensitivity (capacity to detect attCs from the taxon that it was built from), and its specificity (ability to exclude attCs derived from other taxa). CMs that did not achieve a specificity greater than 98% were excluded. This resulted in a set of 11 taxon-specific CMs that proved efficient in the taxonomic assignment of attCs (Supplementary Table 1). The taxa comprised of six Gammaproteobacterial orders (Alteromonadales, Methylococcales, Oceanospirillales, Pseudomonadales, Vibrionales, Xanthomonadales) and an additional five phyla (Acidobacteria, Cyanobacteria, Deltaproteobacteria, Planctomycetes, Spirochaetes). In total, 1978 chromosomal attCs were used to generate the 11 CMs, ranging from 51–505 attCs used for each. Bit score cut-offs for the taxonomic assignment were set for each CM individually in order to

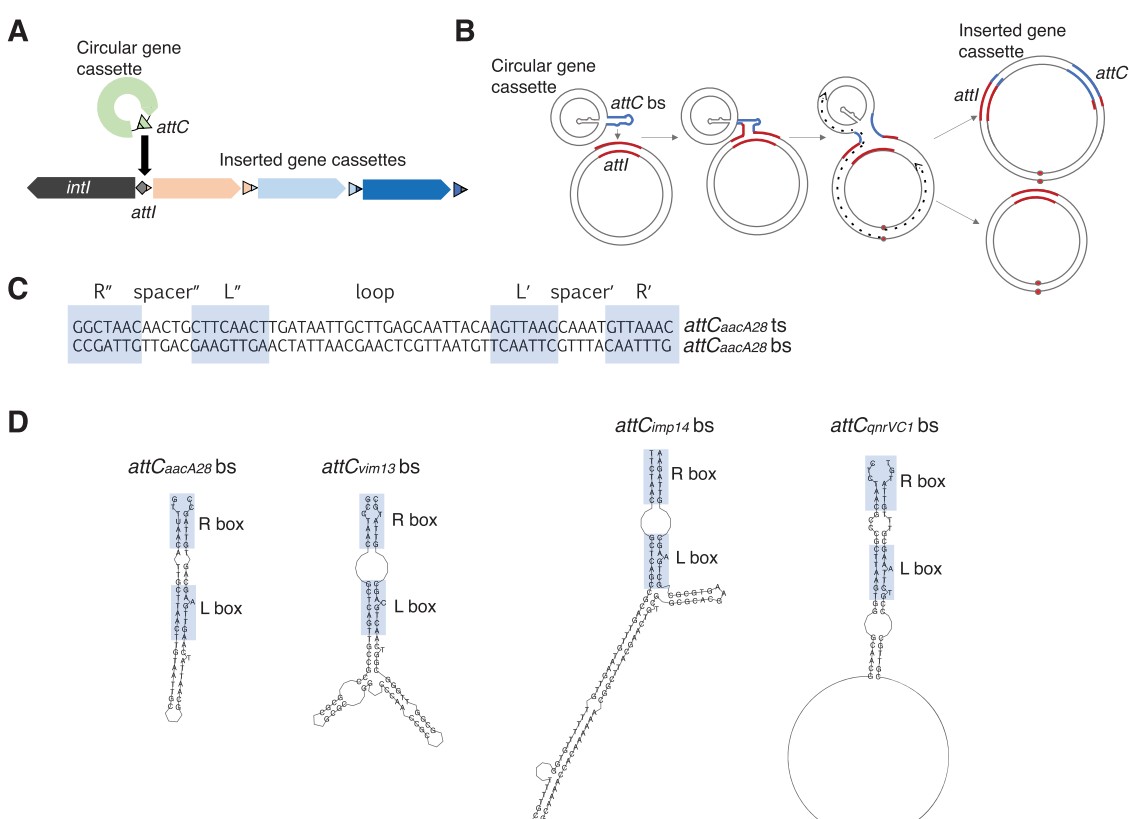

**Fig. 1 The role of *attC* folding structure in gene cassette insertion. A** Integrons carry an integron integrase gene (*intI*) that encodes a tyrosine recombinase (IntI). IntI facilitates the insertion of circular gene cassettes by mediating recombination between the cassette-associated recombination (*attC*) and integron-associated recombination (*attI*) sites. IntI activity can result in arrays of gene cassettes that vary considerably in size (1 to +300). **B** Cassette insertion involves the recombination between *attI* and only the bottom strand of *attC* (*attC* bs). This results in an atypical Holliday junction, which can only be resolved by replication (dotted black arrows; lagging strand not shown). Replication of the recombinogenic strand produces a daughter molecule with the inserted cassette at the *attI* site, while replication of the alternate strand generates the integron without the inserting cassette. **C** The palindromic nature of *attC*s gives rise to their single-stranded folding structure. All *attC* sites have two sets of inverted repeats (R′/R″ and L′/L″), which allow the folding of single-stranded *attC*s. Two spacers, spacer″ and spacer′, separate R″ from L″ and L′ from R′, respectively. The middle region of the *attC* is known as the loop and is highly variable in sequence and size. **D** Shown are the predicted bottom strand folding structures of *attC*s from four antibiotic resistance gene cassettes (*aacA28*, *vim13*, *imp14*, and *qnrVC1*). The variable degree of base-pairing beyond the R and L boxes generates considerable structural diversity among different *attC*s, which in turn impacts their recombination efficiency by different IntIs. Folding structures were predicted by RNAfold v 2.4.16 from the ViennaRNA Package 2.0[40].

maximise sensitivity while ensuring specificity was 98–100% (Supplementary Table 1). The mean specificity for the CMs was 99.6%, ranging from 98.04–100%, and the mean sensitivity was 66.1%, ranging from 30–99.2%. The wide range of CM sensitivities means that the relative number of matches to each taxon cannot be compared, as some have a higher capacity to detect true positives than others. It does, however, indicate that the number of matches to a particular taxon is likely to be a lower-bound estimate.

**Predicting taxonomic origins of resistance gene cassettes from mobile integrons**. Mobile integrons can transfer between species, allowing the acquisition of new gene cassettes from diverse genomic backgrounds[27]. Here, we aimed to predict the original taxonomy of resistance gene cassettes commonly found on mobile integrons using the structural and sequence homology of their *attC* recombination sites.

We used a collection of 108 resistance gene cassettes, reported by Partridge et al.[10]. Using our models, we found that at least 27% of these had *attC*s structurally conserved among one of three taxa (Table 1). On this basis, nineteen resistance cassettes were assigned to Xanthomonadales, six to Spirochaetes, and four to

Vibrionales. It is important to note, however, that each CM exhibits different sensitivities in their ability to detect true positives (Supplementary Table 1), thus the relative contribution of each taxa to the pool of resistance cassettes cannot be compared and all are likely to be lower-bound estimates. Nevertheless, these three taxa have contributed to more than a quarter of resistance cassettes in our dataset, signifying that they are key taxonomic sources of the resistance cassettes now circulating among diverse Gram-negative pathogens.

The types of resistance mechanisms varied between taxa (Table 1). For example, cassettes with Xanthomonadales-type *attC*s encoded a wide range of different resistance proteins, consisting of aminoglycoside (3″) adenylyltransferases, aminoglycoside (2″) adenylyltransferases, aminoglycoside (6′) acetyltransferases, aminoglycoside (3′) acetyltransferases, chloramphenicol acetyltransferases, small drug resistance proteins, a QAC efflux pump, a streptothricin acetyltransferase, and a fosfomycin resistance protein. Whilst, the six cassettes with Spirochaetes-type *attC*s all encoded aminoglycoside (6′) acetyltransferases. Interestingly, cassettes with a predicted Spirochaetes origin represent 30% of all aminoglycoside (6′) acetyltransferases in our dataset, highlighting this phylum as a significant source of this resistance mechanism.

**Table 1 Predicted taxonomic source of resistance gene cassettes found on mobile integrons.**

| Source taxon | Gene cassette | Gene cassette product | Example recipient host from ref. [10] | Accession from ref. [10] |
|---|---|---|---|---|
| Xanthomonadales | aadA1a | Aminoglycoside (3″) adenylyltransferase | Escherichia coli | X12870.1 |
| Xanthomonadales | aadA7 | Aminoglycoside (3″) adenylyltransferase | Escherichia coli | AF224733.1 |
| Xanthomonadales | qacI | Quaternary ammonium compound efflux protein | Escherichia coli | AF205943.1 |
| Xanthomonadales | aacA3 | Aminoglycoside (6′) acetyltransferase | Salmonella enterica | AY123251.1 |
| Xanthomonadales | catB3 | Chloramphenicol acetyltransferase | Enterobacter aerogenes | U13880.2 |
| Xanthomonadales | qacF | Quaternary ammonium compound efflux protein | Enterobacter aerogenes | AF034958.3 |
| Xanthomonadales | aadA6 | Aminoglycoside (3″) adenylyltransferase | Pseudomonas aeruginosa | AF140629.1 |
| Xanthomonadales | aadA24 | Aminoglycoside (3″) adenylyltransferase | Salmonella enteritidis | AM711129.1 |
| Xanthomonadales | aacA37 | Aminoglycoside (6′) acetyltransferase | Pseudomonas aeruginosa | DQ302723.1 |
| Xanthomonadales | catB5 | Chloramphenicol acetyltransferase | Morganella morganii | X82455.1 |
| Xanthomonadales | aadB | Aminoglycoside (2″) adenylyltransferase | Escherichia coli | L06418.4 |
| Xanthomonadales | aadA2 | Aminoglycoside (3″) adenylyltransferase | Escherichia coli | X68227.1 |
| Xanthomonadales | sat2 | Streptothricin acetyltransferase | Escherichia coli | X15995.1 |
| Xanthomonadales | fosE | Fosfomycin resistance protein | Pseudomonas aeruginosa | AY029772.1 |
| Xanthomonadales | aacA7 | Aminoglycoside (6′) acetyltransferase | Enterobacter aerogenes | U13880.2 |
| Xanthomonadales | aacC6 | Aminoglycoside (3) acetyltransferase | Serratia marcescens | AY884051.1 |
| Xanthomonadales | smr2 | Small multidrug resistance protein | Escherichia coli | AY260546.3 |
| Xanthomonadales | aacA29 | Aminoglycoside (6′) acetyltransferase | Uncultured bacterium | AY139599.1 |
| Xanthomonadales | aacC5 | Aminoglycoside (3) acetyltransferase | Vibrio fluvialis | AB114632.1 |
| Spirochaetes | aacA39 | Aminoglycoside (6′) acetyltransferase | Pseudomonas aeruginosa | EU886977.1 |
| Spirochaetes | aacA1:gcuG | Aminoglycoside (6′) acetyltransferase | Escherichia coli | AF047479.2 |
| Spirochaetes | aacA30 | Aminoglycoside (6′) acetyltransferase | Salmonella enterica | AY289608.1 |
| Spirochaetes | aacA17 | Aminoglycoside (6′) acetyltransferase | Klebsiella pneumoniae | AF047556.1 |
| Spirochaetes | aacA16 | Aminoglycoside (6′) acetyltransferase | Citrobacter freundii | Z54241.1 |
| Spirochaetes | aacA28 | Aminoglycoside (6′) acetyltransferase | Pseudomonas aeruginosa | AB104852.1 |
| Vibrionales | dfrA6 | Dihyrofolate reductase | Proteus mirabilis | Z86002.1 |
| Vibrionales | blaP3 | Class A β-lactamase | Pseudomonas aeruginosa | U14749.1 |
| Vibrionales | qnrVC1 | Quinolone resistance protein | Vibrio cholerae | EU436855.2 |
| Vibrionales | blaP7 | Class A β-lactamase | Vibrio cholerae | AF409092.1 |

To further validate our findings, we searched for examples of the complete cassettes in the predicted taxon of origin. To do this, we searched for the cassette open reading frames (ORFs) among all complete bacterial chromosomes available in NCBI (n = 24,143 as of 27th January 2021). We found that three of the resistance cassette ORFs were present in bacterial chromosomes, and in all cases, they were part of gene cassettes. All three were only found in the taxa predicted by our CMs and were only present once each among all 24,143 chromosomes. All three were part of gene cassettes within SCIs (100% nucleotide identity

spanning the ORF and *attC* site). Two of the cassettes, which we had predicted to originate from Xanthomonadales, encoded a chloramphenicol acetyltransferase (*catB3*) and a QAC efflux pump (*qacI*). Interestingly, both of these were found in the same chromosomal integron of *Lysobacter oculi* (Accession: CP029556.1), part of the order Xanthomonadales. The third cassette encoded a quinolone resistance protein (*qnrVC1*) that we predicted to have a Vibrionales origin, and was indeed found within a cassette array of *Vibrio alginolyticus* (Accession: CP060386.1). The fact that all of these ORFs were found in the

taxa predicted by our CMs, and were not present in any other taxa, strongly supports the predictions made by our CMs. In addition, the *attC* of the *blaP3* resistance cassette, which our modelling predicted to have originated from Vibrionales has previously been reported to share overall sequence similarity of 90% with *Vibrio cholerae attC* sequences[28].

Knowledge of the taxonomic origins of resistance gene cassettes, and what ecological and physiological traits are shared by these taxa, can allow us to predict environmental hotspots or conditions that contribute to the emergence of novel resistance genes[26]. The origins of up to 30 antibiotic resistance genes have been previously proposed, however, none of these attributions included integron gene cassettes[26]. Predicting chromosomal origins of resistance genes generally involves examining the genes in the immediate vicinity that have been co-mobilised[26]. However, this approach cannot work for integron gene cassettes, as each cassette is a modular mobile unit, thus, co-mobilisation and continuous maintenance of the same arrangement of multiple cassettes is extremely unlikely. However, by using the taxonomic signatures preserved in the structure and sequence of their recombination sites, we are able to predict from which taxa cassettes have likely originated.

### Environmental clustering of *attC*s: predicting environments of origin of resistance cassettes.

Several studies have shown that the phylogeny of integron integrases (IntI) cluster according to the host environment, with marine IntIs forming one major clade and soil/freshwater IntIs forming another[4,6,29]. The inverted IntIs form a sub-clade within the larger soil/freshwater clade[29]. We therefore investigated whether *attC*s exhibit similar environmentally explicit clustering. To do this, we used ten representative *attC*s from each taxon-specific CM, which together with the 108 resistance cassettes from mobile integrons[10], were clustered based on sequence and folding structure.

We found that *attC*s cluster into three major clades (Fig. 2). One that is distinctly a marine clade, the second is a soil/freshwater clade, and a third clade that we have labelled as 'Xanthomonadales-like' (XL), as only *attC*s from the Xanthomonadales CM fell into this clade. These distinct clusters allow us to infer from which environments resistance cassettes have likely originated. The XL clade included the greatest number of resistance cassette *attC*s and encompassed the greatest range of resistance types (Fig. 2). However, the majority of β-lactamases (52.9%) and aminoglycoside acetyltransferases (54.2%), the two most common types of antibiotic resistance cassettes[10], formed part of the marine clade. We therefore, for the first time, highlight the marine environment as a key source of these two prevalent types of integron-mediated resistance.

To further investigate if any other taxa carry Xanthomonadales-like *attC*s, we built a CM using the 58 *attC*s from the Xanthomonadales-like clade (Fig. 2) and used this to search all 24,143 bacterial chromosomes available in NCBI. We detected 941 Xanthomonadales-like *attC*s within 56 genomes (Supplementary Table 2). These were almost all found in Xanthomonadales, with 935 *attC*s (99.4%) from 54 Xanthomonadales genomes. Among them were the genera *Xanthomonas*, *Lysobacter*, *Luteimonas*, *Pseudoxanthomonas*, *Thermomonas,* and *Stenotrophomonas* (Supplementary Table 3). Most of these genera are commonly found associated with plant leaves and roots[30–33], suggesting that the Xanthomonadales-like *attC* clade might represent plant-associated environments. If this is the case, then plant-associated bacteria might be a significant source of a large proportion of resistance gene cassettes. Interestingly, the class 1 integron platform is also thought to have originated from plant leaf surfaces[9]. However,

since not all of these Xanthomonadales genera are endemic to plants and since we lack additional taxa outside of Xanthomonadales, and thus cannot distinguish if this clade represents an environmental or taxonomic grouping, further evidence would be required to support this.

### Convergent and co-evolution of *attC* sites and integron integrases.

Here, we show that *attC*s cluster according to the host environment, with a clear distinction between marine *attC*s and soil/freshwater *attC*s. The same pattern has been previously shown for the phylogeny of integron integrases[4,6,29]. Using IntI protein sequences extracted from genomes used in the present study, we also show that their phylogeny clusters according to the host environment (Supplementary Figure 2). We propose that this shared clustering pattern is the result of convergent and co-evolutionary processes.

Cassette insertion and excision are driven by IntI-mediated *attC* x *attI* and *attC* x *attC* recombination, respectively. The efficiency of these recombinations largely depends on the folding structure of *attC*s[18] (Fig. 1D). However, the recombination efficiency over a range of diverse *attC* sites varies between different integron integrases. For example, the sedentary *Vibrio cholerae* IntIA can recognise a narrower range of *attC*s than those recognised by the mobile class 1 integron integrase[21]. Thus, within a bacterial taxon, the degree of divergence in *attC* structures will likely predict the *attC* specificity of the endogenous integrase. As the *intI* gene evolves, the *attC* folding structures will likely co-evolve to maintain functionality, and vice versa. Additionally, cassette genesis will also influence the degree of divergence among *attC*s. Given the variability in *attC* homogeneity within a taxon, it is possible that some species might generate highly similar *attC*s, while other species might generate variable *attC*s. In either scenario, however, the native mechanism of cassette genesis, along with the selection strength that controls *attC* divergence, and the specificity range of the endogenous integrase are all intrinsically linked. Co-evolution of these three processes would ensure the maintenance of integron functionality and recombination efficiency.

Further, horizontal transfer of cassettes is likely to be more prevalent among bacteria inhabiting the same environment. Here, selection will favour integrons that can recognise and incorporate foreign cassettes with novel functions. This is strongly supported by the observed environmental clustering of *attC* structures and IntI phylogeny. Further, the clustering of IntIs and *attC*s are incongruent with the phylogenetic clustering of their host organisms[34]. Thus, taxa that are not necessarily closely related by descent, have more similar *intI* genes and *attC* structures. This is evidence of convergent evolution, allowing IntI recombinatorial activity on structurally similar *attC* substrates that are more prevalent in the same environment. A selective advantage would be gained by integron platforms that can successfully recognise and incorporate foreign cassettes from the local environment.

### Conclusion

In the present study, we show for the first time that the taxonomic origins of integron gene cassettes can be predicted from the structure and sequence of their *attC* recombination sites. Using this approach, we predicted the source taxon of 29 out of 108 resistance gene cassettes spread by mobile integrons. These appear to have originated from three taxa (Xanthomonadales, Spirochaetes, and Vibrionales). We searched for evidence of these cassettes residing in their presumptive ancestral hosts. We found three of these cassettes in chromosomal integrons in the predicted taxon, and these cassettes were not present in any other bacterial

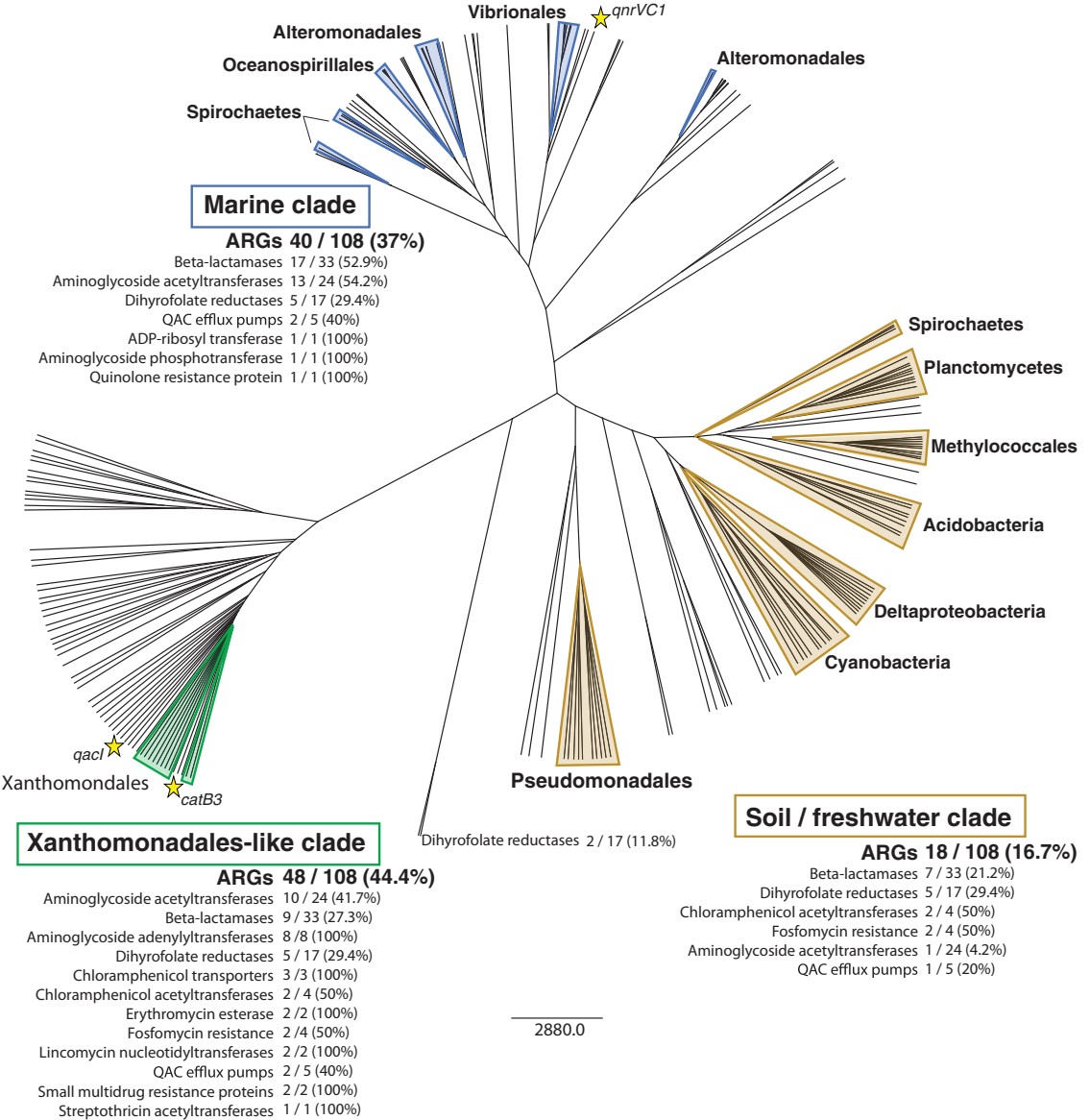

**Fig. 2 Structure-based clustering of *attC*s.** Ten top-scoring (based on covariance model (CM) bit scores) *attC*s for each taxon-specific CM are outlined by shaded triangles. Each non-shaded branch represents an *attC* site from one of 108 different resistance gene cassettes annotated by Partridge et al.[10]. The structures of *attC*s cluster according to host environment, forming three major clades. For each clade, the abundance of each resistance type is displayed, along with its relative proportion among all cassettes of that resistance type. Branches with yellow stars represent *attC*s from the three gene cassettes observed residing in the sedentary chromosomal integrons of the ancestral taxa predicted by our covariance models. We predicted that two of these (*qacI* and *catB3*) originated in Xanthomonadales, and the third (*qnrVC1*) in Vibrionales. Relative distances show all three are highly similar to their predicted taxa, further validating these predictions. See Supplementary Figure 1 for a tree with all branches labelled.

chromosome. These findings support our *attC* taxonomy predictions. We also show that *attC*s, based on structure and sequence, cluster according to the host environment rather than host phylogeny. We use this clustering to predict the source environments of each resistance cassette. Together, our findings represent the first evidence of the taxonomic and environmental origins of resistance cassettes.

The search for and use of novel antibiotics will inevitably result in the evolution and spread of novel resistance genes among pathogenic bacteria. Mobile integrons, will most likely play a role in their dissemination. Knowledge of the origins of already problematic resistance genes can allow us to predict hotspots for the emergence of the next generation of resistance genes before they enter clinical settings, and establish mechanisms to prevent them from doing so.

## Methods

**Collecting *attC*s from genomic sequences.** Our approach to detect *attC* recombination sites was based on methods developed by Pereira et al.[35]. This first involved using HattCI v1.0b[36], which uses a generalised hidden Markov model to detect the nucleotide sequence of each core motif of an *attC* site (i.e. R″ - spacer″ - L″ - loop″ - L′ - spacer′ - R′[37] (Fig. 1C)). HattCI was implemented so that both strands were analysed in batches of 40 sequences [parameters: -b -s 40 -t 8]. All output sequences generated by HattCI were then screened for the conserved *attC* folding structure. The consensus structure was generated using a structural alignment of 231 manually curated *attC*s, largely from class 1 and 2 integron cassette arrays, available in Supplementary Material Section B of Pereira et al.[36]. The structural alignments were generated using LocARNA v1.9.2.1[38–40], which uses tools built on the Turner free energy model to simultaneously fold and align input sequences. The alignments of the *attC*s were anchored so that all complementary conserved motifs were forced to align (i.e. R″, L″, L′, R′ (Fig. 1C)) [parameters: mlocarna --anchor-constraints --stockholm --threads 8]. A covariance model (CM) was built from the structural alignment using the cmbuild and cmcalibrate tools in the Infernal v1.1.2 package[41] with default parameters. The CM was used to screen the HattCI output for the correct folding structure necessary

for *attC* functionality using Infernal's cmsearch tool with an E-value threshold of 0.01. Putative *attC*s were subject to further filtering to remove singletons, with the retained set being those that were clustered (at least two *attC*s) with no more than 4 kb between each[5], that being twice the size of the largest annotated gene cassette[42].

To collect *attC*s from distinct bacterial taxa, we first applied the above approach to screen an initial batch of complete genomes known to carry integrons[5]. We excluded any sequences that represented plasmids, as these would consist of *attC*s in cassettes from mobile integrons, which could have been acquired from a diverse set of taxa. The initial batch consisted of 1,825 complete chromosomal sequences, representing 20 bacterial phyla. We collected *attC*s for each phlyum separately, and redundancy was removed from each set of *attC*s by discarding identical sequences with CD-HIT v4.6[43,44] [parameters: cdhit-est -c 1.0 -aL 1.0]. We split the *attC*s from Gammaproteobacteria into order-level groupings due to the extensive number of *attC*s recovered and sequenced genomes present in this group. For those taxa in which we detected less than 50 non-redundant *attC*s, we sought additional chromosomal sequences from the NCBI Assembly database [accessed December 2020]. Any taxon that still had less than 50 representative *attC*s were excluded from further analysis.

**Creating taxon-specific *attC* covariance models (CMs)**. For each taxon-specific set of *attC*s, we created a CM based on a structural alignment using methods described above. We then tested the efficacy of the models in predicting the host taxonomy of provided *attC* sequences. To do this, we searched each taxon-specific CM against our complete set of *attC*s (n = 2,352), using the cmsearch tool in the Infernal v1.1.2 package [parameters: --cpu 8 --notrunc --nohmm]. The efficacy of each CM was determined by its sensitivity, that being its capacity to detect *attC*s from the particular taxon that it was built from (true positives), and its specificity, which is the ability to exclude *attC*s derived from other taxa (false positives). CMs that did not achieve a specificity greater than 98% were excluded. All CMs are available as Supplementary Data 1.

**Assigning ancestral host taxonomy to antibiotic resistance gene cassettes**. CMs that passed the efficacy screening were used to assign host taxonomy to a set of 108 resistance gene cassettes from mobile integrons, obtained from Table 1 in Partridge et al.[10]. The CMs were searched against the *attC*s from each resistance cassette using Infernal's cmsearch as described above. The bit score cut-offs for the taxonomic assignment were set for each CM individually in order to maximise sensitivity while ensuring specificity was 98–100% (See Supplementary Table 1 for specific bit scores).

In addition, we sought to determine if any of the resistance cassettes of the mobile integrons could be observed in their putative ancestral chromosomal context. ORFs of the resistance cassettes that could be assigned a taxonomy were aligned against all 24,143 complete bacterial chromosomes available in NCBI [downloaded on 27 January 2021]. Multiple alignments were implemented using BLAST v2.7.1[45] with 98% identity and 100% query cover thresholds [parameters: -task megablast -perc_identity 98 -qcov_hsp_perc 100 -num_threads 8]. Each hit was manually checked to exclude all instances of mobile integrons that had transposed into chromosomes as these are unlikely to be the ancestral sources.

We applied a structural-based clustering approach to visualise how similar *attC*s of the resistance cassettes were with representative *attC*s from each taxon. For this, we used RNAclust v1.3[46], which builds on locARNA to create a hierarchical-clustering tree from a WPGMA analysis. We collected the top 10 representative *attC*s for each CM based on their bit scores. These, along with the resistance cassette *attC*s, were clustered using RNAclust's default parameters.

**Phylogenetic analysis of integron integrases**. Integron integrases (IntIs) were obtained from selected genomes that had the top-scoring *attC*s for each CM. We aimed to collect five representative IntI sequences from each taxon using a profile HMM provided by Cury et al.[5]. The profile HMM was based on the additional domain that is unique to integron integrases, separating them from other tyrosine recombinases[47]. For each selected genome, proteins were annotated using Prodigal v2.6.3[48] and the profile HMM was used to determine which sequences represented IntIs with the hmmsearch tool from the HMMER v3.2 package[49]. Any partial IntI sequences were discarded from the phylogenetic analysis and the remainder were manually screened to ensure that they possessed the complete additional domain. The orientation of the *intI* gene in relation to the cassette array was confirmed using IntegronFinder v1.5.1[5] [parameters: --local_max --func_anot].

IntI sequences were aligned using MAFFT v7.271[50] [parameters: --localpair --maxiterate 1000]. The best substitution model to suit the alignment was determined using ModelFinder[51], which was used to construct a maximum-likelihood tree with IQ-TREE v1.6.12[52,53] with 1,000 bootstrap replicates [parameters: -m MFP -alrt 1000 -bb 1000 -nt 8].

**Reporting summary**. Further information on research design is available in the Nature Research Reporting Summary linked to this article.

## Data availability

Specific bacterial genomes were downloaded from NCBI via accessions provided by Cury et al.[5] using the following Perl command [perl -e 'use LWP::Simple;getstore("http://eutils.ncbi. nlm.nih.gov/entrez/eutils/efetch.fcgi?db=nucleotide&rettype=fasta&retmode=text &id="join(",",qw(<space-separated list of nucleotide accession numbers)),"output_filename. fasta");']. The remaining genome sequences were downloaded directly from the NCBI Assembly database (https://www.ncbi.nlm.nih.gov/assembly).

## Code availability

Software used in this study are LocARNA v1.9.2.1; HattCI v1.0b; Infernal v1.1.2; CD-HIT v4.6; BLAST v2.7.1; RNAclust v1.3; Prodigal v2.6.3; HMMER v3.2; IntegronFinder v1.5.1; MAFFT v7.271; IQ-TREE v1.6.12; ViennaRNA v2.0. Specific parameters used for each software are provided in detail in the Methods section.

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

## Acknowledgements

This research was supported by the Australian Research Council Discovery Grant DP200101874. TMG would like to thank Mary and Saoirse Ghaly for loving support.

## Author contributions

TMG contributed to the conception of the study, performed all data analyses, wrote the original draft of the paper, and contributed to the final editing and revision of the paper. SGT contributed to the conception of the study and the final editing and revision of the paper. MRG contributed to the conception of the study and the final editing and revision of the paper. All authors contributed to the article and approved the final submitted version.

## Competing interests

The authors declare no competing interests.
