## [Peer Review File · Communications Biology]

Reviewers' Comments:

Reviewer #1:

Remarks to the Author:

Comments about the paper from Timothy M Ghaly, Sasha G Tetu and Michael R. Gillings:
Predicting the origins and environmental sources of integron gene cassettes using structural and sequence homology of attC sites.

Comments for the editor:

In this paper, the authors use attC sites (structure and sequence) to predict the source of antibiotic resistance gene cassettes carried by Mobile Integrons, now largely disseminated among bacteria. They found that at least 26% of attC sites are conserved among several taxa and that some cassettes were still residing in SCIs. Interestingly, they show that cassettes cluster much more according to their host environment than according to the bacterial phylogeny. Moreover, they propose a co-evolution between integrases and attC sites.

The data collected in this paper are of interest for people studying integrons but also bacterial evolution. The question posed by the authors is well defined and the assays developed are very well described and suitable. The paper is clearly written. I think that the manuscript could adhere to the relevant standards for reporting and data deposition in Communications biology. I suggest major and minor concerns.

Major concern:

1) My major revision concerns the employed term of origin of gene cassettes. For me, this term is confusing and not appropriated, the term "sources of cassettes" is much more suited and should be used all along the ms. Indeed, using the term "origin of gene cassettes" could be mistaken with the cassette genesis;

2) In the abstract, the authors could precise that they "found resistance still residing in SCI chromosome".

3) The authors can also discuss of the previous observation made by the mazel lab Mazel et al, 1998, Science. "We found a 90% sequence identity between the VCR sequences and the 59-base element associated with blaP3, which is an integron-associated antibiotic resistance gene encoding the carbenicillinase CARB-4 isolated from Pseudomonas." These previous results are completely in accord with the results obtained here by the authors and could be mentioned in the introduction part of the ms.

4) The authors argue that there is a co-evolution between integrases and attC sites and that integrases do not recombine all the attC sites and present preferential attC sites i.e. the authors cite the integrase IntIA. Indeed, in the paper of Biskri et al, 2005, it is demonstrated that IntIA recombine a narrower attC sites than other MI integrases. The authors could attempt to explain results obtained in this paper using their analysis? Indeed, it was demonstrated, by in vivo recombination assay, that IntIA recombines better both attCaadA7 and attCaadAB sites than attCorfA, attCereA and attCdfrA1. Are both sites attCaadA7 and attCaadAB present in vibronaceae?? Explaining thus a better efficiency of recombination due to their co-evolution with intIA??

Minor revisions:

1) Line 71: change "after the typical second-strand exchange" by "by the typical second-strand exchange" and "after replication" by "by replication"

2) Figure 1A and B: the authors could attempt to homogenize both Figure 1A and 1B, by example by using the same colours for the attI site and for gene cassettes and attC sites.

Reviewer #2:

Remarks to the Author:

In the manuscript "Predicting the origins and environmental sources of integron gene cassettes using structural and sequence homology of attC sites" Ghaly et al. recover almost 2000 attC sites from sedentary chromosomal integrons to model the structure and sequence homology allowing to

detect signatures that relate cassettes to the bacteria where they are naturally found (their origin). They are able to distinguish 3 large groups of attC sites. They then use this to predict the origin of cassettes found on mobile integrons.

I believe this manuscript is interesting, relevant in the field of integrons and antimicrobial resistance, and represents a relevant increase of our understanding of the ecology of integron cassettes. The manuscript is clear, well written and based on a solid knowledge on integrons. I have only minor format comments and a couple of comments that are more of a discussion.

Minor comments:

Sentence in line 173 is difficult to read. Could you rephrase it?

The authors collect attCs using HattCI instead of IntegronFinder. Not being a bioinformatician I would have thought that IntegronFinder is the new gold standard in the field. Could the authors explain the pro's and cons of choosing HattCI?

Could the authors provide an exhaustive version of table 1? I think it would represent a valuable asset for readers to be able to assess directly the origin of a given cassette in Partridge's list.

I would also think readers would like to see a figure similar to Figure 2 but with the 2000 attCs from SCIs. Then it would make even more sense to keep the ten best scoring attCs for comparison with ARGs in Figure 2

Figure 2 could benefit from some clarifications. Because it is a very informative figure, the legend should describe it more extensively. For instance, percentages might be more readily understandable if the proportion is shown. When one sees in the Xanthomonadales "Beta-lactamases 9 (27.3)" the first reflex can be to assume that it is 9/48 and that you are representing the importance of that resistance mechanism within Xanthomonadales, while instead, you are representing the percentage of all betalactamases in this clade (9 out of 33 found in the total 108 ARGs) so maybe putting it as "betalactamases 9/33 (27,3%)" is more intuitive.

In accordance to my previous request on providing an exhaustive version of Table 1, it would be very interesting that the authors label the branches from ARGs attCs in figure 2. I understand this can turn figure 2 into a nightmare, so I suggest they split it in three supp figures (one for each taxon) in which they label branches so that we can clearly see to what attC site each branch refers. This provides the reader with a high definition view of the origin of cassettes that would add value to the author's results (for example, apart from being originated in the marine clade, I could see that a cassette of interest originates in Alteromonadales,)

Discussion:

There are some points where I have a somewhat different perception than the author's, but I would like to maintain their freedom to take my comments as a discussion and use them to enrich their text if they feel they are useful.

Lines 309-314 and 324-333. I am not sure there is a co-evolutionary process going on between integrases and attC sites. There is obviously a selective force to maintain their interaction, but the high identity of cassettes in SCIs could stem from different processes. It has been interpreted by others as representative of a mechanism of cassette genesis within the genome. In this case the grouping of attCs in the same clades as integrases would be the consequence of a cassette generator that, as the integrase, would also follow the branching and, in this evolutionary divergence, would change the sequence of the attC's it generates. As you also mention, another option would be that cassettes are exchanged by bacteria sharing the same environment i.e: HGT. This would assimilate better to a co-evolution but, because both scenarios (cassette generators and HGT) are not incompatible, it could still be due to the sequence specificity of cassette generators. Is there anything in the data generated in this paper that can guide the reader between these options?

About the specificity of integrases

Any idea from your tree on whether the broad specificity of IntI1 can be inferred as a broader range of attCs in bacteria that have close homologs to IntI1 as part of their SCI?

Please let me reinstate my overall satisfaction with the paper. I really believe the authors provide here a novel approach to understand the ecology of cassettes that makes a lot of sense.

Reviewer #3:

Remarks to the Author:

The paper by Ghaly et al describes a study modelling the evolutionary sequence and structure conservation of integron attC sites across taxa and environments. The authors find that 1) taxonomic origins of integron gene cassettes can be predicted from the structure and sequence of their attC sites, and 2) attC sites cluster according to host environment rather than host phylogeny. They use these two findings to predict the origins for 28 antibiotic resistance cassettes spread by mobile integrons and predict the source environments of resistance cassettes. Both of these findings are new and reflect a novel approach to identify the origins of antibiotic resistance genes (cf. for example Ebmeyer et al in *Communications Biology*, which the authors cite). Since the origins for resistance genes are interesting both from a theoretical standpoint and for potential mitigation of future resistance threats, this is of great interest to the field.

The methods are sound and reproducible. I usually have criticisms about bioinformatic methods, but I could not find any critical flaws in the authors' approach. That said, I do wonder why the authors have chosen to make assignments using e-values (which is dependent on database size) rather than score or score per length? Is there any good reason for this? Note that using e-values is not scalable to big datasets.

However, I do find the manuscript a bit hard to follow. I don't feel that there is always a logical flow between the different paragraphs. It is hard to be specific, but I struggled a bit to get the pieces together.

I am also not entirely convinced by the method the authors use to connect environments to the origins of certain resistance gene classes. This analysis could be made more robust by also adding in, for example, metagenomic data, and by covering a larger range of environments. The categories "marine", "freshwater/soil" (which by the way is incredibly broad) and "Xanthomonadales-like" do not at all consider host-associated environments, for example. While I think that the reasoning around taxonomic origins is sound and robust, I feel that this approach lacks the support that the authors think they have to make claims about source environments.

Aside from this, I only have a few minor comments:

- Line 28: Unsure what the authors are implying with "still" here. Is that surprising?
- Line 31: Not necessarily sources, could also be the environment the cassette provides the most advantage in.
- Line 104-108: This is not clear until after reading the rest of the paper. Perhaps this could be rephrased?
- Line 206: Likely to what? This sentence seems incomplete.
- Line 225: As noted later by the authors, this is a substantial under-prediction. I would like the authors to comment a bit further on the implications of this under-prediction.
- Line 264: Very minor comment which is not that important for the paper, but how do the authors define emergence? All of these resistance genes seem to have been pre-existing, so what is the emergence event, as defined by the authors of this paper?
- Line 329-330: I don't think that the authors do any effort to test whether taxa that are not closely related by descent have more similar intI genes and attC structures, and I think they should. A formal test of that would strengthen the conclusions of the paper.
- Line 353-354: Can the knowledge the authors generate really allow us to predict hotspots for the emergence of the next generation of resistance genes, if the origin is as wide as "the marine

environment"? Can the authors connect this to any credible intervention? How does this knowledge make us more likely to be able to mitigate future resistance development?

Reviewer #1 (Remarks to the Author):

Comments about the paper from Timothy M Ghaly, Sasha G Tetu and Michael R. Gillings: Predicting the origins and environmental sources of integron gene cassettes using structural and sequence homology of attC sites.

Comments for the editor:

In this paper, the authors use attC sites (structure and sequence) to predict the source of antibiotic resistance gene cassettes carried by Mobile Integrons, now largely disseminated among bacteria. They found that at least 26% of attC sites are conserved among several taxa and that some cassettes were still residing in SCIs. Interestingly, they show that cassettes cluster much more according to their host environment than according to the bacterial phylogeny. Moreover, they propose a co-evolution between integrases and attC sites.

The data collected in this paper are of interest for people studying integrons but also bacterial evolution. The question posed by the authors is well defined and the assays developed are very well described and suitable. The paper is clearly written. I think that the manuscript could adhere to the relevant standards for reporting and data deposition in Communications biology. I suggest major and minor concerns.

Major concern:

1) My major revision concerns the employed term of origin of gene cassettes. For me, this term is confusing and not appropriated, the term “sources of cassettes” is much more suited and should be used all along the ms. Indeed, using the term “origin of gene cassettes” could be mistaken with the cassette genesis;

>>>>> As suggested, we have altered the term “origin” to “source” in most cases throughout the manuscript, including the title. We have left the term “taxonomic origin” as we believe this helps clarify that we are not referring to cassette genesis. To further avoid any confusion, we have included the line “Hereafter, we use the term ‘origin’ to represent the SCI source from which the cassettes were acquired by mobile integrons” in the introduction (lines: 90-91)

2) In the abstract, the authors could precise that they “found resistance still residing in SCI chromosome”.

>>>>> As suggested by the reviewer, we have clarified the abstract to reflect that the resistance cassettes were part of SCIs (line 28).

3) The authors can also discuss of the previous observation made by the mazel lab Mazel et al, 1998, Science. “We found a 90% sequence identity between the VCR sequences and the 59-base element associated with blaP3, which is an integron-associated antibiotic resistance gene encoding the carbenicillinase CARB-4 isolated from Pseudomonas.” These previous results are completely in accord with the results obtained here by the authors and could be mentioned in the introduction part of the ms.

>>>>> We thank the Reviewer for pointing this out. We have included this in the Results and Discussion section (Lines 269-272). Indeed, this previous result is in accord with our findings, particularly because our models predicted this *blaP3* cassette to have originated from a Virbrionales SCI.

4) The authors argue that there is a co-evolution between integrases and attC sites and that integrases do not recombine all the attC sites and present preferential attC sites i.e. the

authors cite the integrase IntIA. Indeed, in the paper of Biskri et al, 2005, it is demonstrated that IntIA recombine a narrower attC sites than other MI integrases. The authors could attempt to explain results obtained in this paper using their analysis? Indeed, it was demonstrated, by in vivo recombination assay, that IntIA recombines better both attCaadA7 and attCaadAB sites than attCorfA, attCereA and attCdfrA1. Are both sites attCaadA7 and attCaadAB present in vibrionaceae?? Explaining thus a better efficiency of recombination due to their co-evolution with intIA??

>>>> Based on our reanalysis using CM bit scores instead of E-values (as suggested by Reviewer 3), both attC_{aadB} and attC_{aadA7} were assigned to Xanthomonadales. Similarly, according to the structure-based tree, they both cluster tightly with Xanthomonadales attCs and not with Vibrionales. To further investigate the point raised by the Reviewer, recombination efficiency of these cassettes should be compared using IntIA vs a Xanthomonadales IntI.

Indeed, this is an interesting point raised by the Reviewer, however, we have limited data on the recombination activity of different IntIs with different attC substrates. Ideally, recombination assays could be done with these resistance cassettes to confirm if recombination efficiency is greatest with integrases from the predicted taxa. We would predict a better recombination efficiency mediated by IntIs that have co-evolved with the attC substrate. However, this is beyond the scope of the current study.

Minor revisions:

1) Line 71: change “after the typical second-strand exchange” by ‘by the typical second-strand exchange” and “after replication” by “by replication”

>>>> This has been corrected as recommended

2) Figure 1A and B: the authors could attempt to homogenize both Figure 1A and 1B, by example by using the same colours for the attI site and for gene cassettes and attC sites.

>>>> Although this is a good point raised by the Reviewer, we feel that this would subtract from the main components we are trying to illustrate in either figure. For example, in Fig. 1A, the attCs are coloured the same as each gene cassette to show the splitting of an attC and the linearisation of a cassette after insertion. In Fig. 1B, we are highlighting attC x attI recombination and how this involves a folded attC structure, thus we want to only have these two components coloured. The different key components of these two figures means that we would prefer to have different colour schemes.

Reviewer #2 (Remarks to the Author):

In the manuscript “Predicting the origins and environmental sources of integron gene cassettes using structural and sequence homology of attC sites” Ghaly et al. recover almost 2000 attC sites from sedentary chromosomal integrons to model the structure and sequence homology allowing to detect signatures that relate cassettes to the bacteria where they are naturally found (their origin). They are able to distinguish 3 large groups of attC sites. They then use this to predict the origin of cassettes found on mobile integrons.

I believe this manuscript is interesting, relevant in the field of integrons and antimicrobial resistance, and represents a relevant increase of our understanding of the ecology of integron cassettes. The manuscript is clear, well written and based on a solid knowledge on integrons. I have only minor format comments and a couple of comments that are more of a discussion.

Minor comments:

Sentence in line 173 is difficult to read. Could you rephrase it?

>>>> As suggested by the Reviewer, we have rephrased this sentence (lines 174-175).

The authors collect attCs using HattCI instead of IntegronFinder. Not being a bioinformatician I would have thought that IntegronFinder is the new gold standard in the field. Could the authors explain the pro's and cons of choosing HattCI?

>>>> IntegronFinder is an extremely useful tool, and indeed we do use it to determine the orientation of the IntIs that we extracted. However, we decided to use a two-step approach for detecting attCs instead of IntegronFinder. First, we used HattCI, which looks for the individual states of an attC (R", spacer", L", loop, L', spacer', R'), and then we use Infernal for structural validation of sequences that have those states. IntegronFinder only uses the Infernal step to detect attCs. It takes genomic or metagenomic sequence as input, and uses Infernal to screen for attC folding structures using a universal attC covariance model.

The two-step approach we've used has two benefits. First, it is more stringent than using IntegronFinder, and thus provides higher confidence in the results. The second benefit is that it is much quicker and possibly more sensitive at finding attCs. The reason for this is that when you use Infernal by itself, like IntegronFinder does, it is extremely slow for large datasets. On the other hand, HattCI is extremely fast and by using the output of HattCI as the input for Infernal, we can speed up the process drastically by reducing the sequence space that Infernal needs to search through.

Could the authors provide an exhaustive version of table 1? I think it would represent a valuable asset for readers to be able to assess directly the origin of a given cassette in Partridge's list.

>>>> As suggested by the Reviewer, we have now added further information to Table 1 to more easily cross-reference Partridge's table.

I would also think readers would like to see a figure similar to Figure 2 but with the 2000 attCs from SCIs. Then it would make even more sense to keep the ten best scoring attCs for comparison with ARGs in Figure 2

>>>> Although this would be ideal, it would not be computationally possible. Unlike a phylogenetic tree, which is based on a single sequence alignment of all input sequences, a structural-based tree involves the simultaneous folding and aligning of all sequences computed by individual pairwise comparisons. Unfortunately, with existing methods, we would not be able to compute such a task for all 2000 attCs. For that reason, we selected the then best-scoring attCs to represent each taxon.

Figure 2 could benefit from some clarifications. Because it is a very informative figure, the legend should describe it more extensively. For instance, percentages might be more readily understandable if the proportion is shown. When one sees in the Xanthomonadales "Beta-lactamases 9 (27.3)" the first reflex can be to assume that it is 9/48 and that you are representing the importance of that resistance mechanism within Xanthomonadales, while instead, you are representing the percentage of all betalactamases in this clade (9 out of 33 found in the total 108 ARGs) so maybe putting it as "betalactamases 9/33 (27,3%)" is more intuitive.

>>>> We thank you the Reviewer for this suggestion. We have now included the proportions and percentage next to each type of resistance gene. We agree, that this makes the stats more easily understandable. We have also edited the figure legend to make this clearer .

In accordance to my previous request on providing an exhaustive version of Table 1, it would be very interesting that the authors label the branches from ARGs attCs in figure 2. I understand this can turn figure 2 into a nightmare, so I suggest they split it in three supplementary figures (one for each taxon) in which they label branches so that we can clearly see to what attC site each branch refers. This provides the reader with a high definition view of the origin of cassettes that would add value to the author's results (for example, apart from being originated in the marine clade, I could see that a cassette of interest originates in Alteromonadales,)

>>>> As suggested by the Reviewer, we have now included an additional supplementary figure showing the labels for all branches of the attC tree.

Discussion:

There are some points where I have a somewhat different perception than the author's, but I would like to maintain their freedom to take my comments as a discussion and use them to enrich their text if they feel they are useful.

Lines 309-314 and 324-333. I am not sure there is a co-evolutionary process going on between integrases and attC sites. There is obviously a selective force to maintain their interaction, but the high identity of cassettes in SCIs could stem from different processes. It has been interpreted by others as representative of a mechanism of cassette genesis within the genome. In this case the grouping of attCs in the same clades as integrases would be the consequence of a cassette generator that, as the integrase, would also follow the branching and, in this evolutionary divergence, would change the sequence of the attC's it generates. As you also mention, another option would be that cassettes are exchanged by bacteria sharing the same environment i.e: HGT. This would assimilate better to a co-evolution but, because both scenarios (cassette generators and HGT) are not incompatible, it could still be due to the sequence specificity of cassette generators. Is there anything in the data generated in this paper that can guide the reader between these options?

>>>>> This is a very good point raised by the Reviewer. Indeed, we have now included discussion of this in the manuscript (lines 334-340). We also agree that these two scenarios are not separate but are linked. We would still argue that co-evolution between IntIs and attCs is an essential process to maintain functionality. Given the variability in attC homogeneity within a taxon, it is possible that some species might generate highly similar attCs, while other species might generate variable attCs. Either way, the native mechanism of cassette genesis, along with the selection strength that controls attC divergence, and the specificity range of the endogenous IntI are intrinsically linked. Co-evolution of these three processes would ensure the maintenance of integron functionality and recombination efficiency.

About the specificity of integrases

Any idea from your tree on whether the broad specificity of IntI1 can be inferred as a broader range of attCs in bacteria that have close homologs to IntI1 as part of their SCI?

>>>> This is a really interesting idea. Though we simply don't know the specificity range of IntIs (excepting IntI1,2,3 and IntIA). We also don't know to what degree does

sequence similarity influence the range of attCs that are recognised by an IntI (i.e. a single amino acid difference at a particular residue might make a larger difference than several amino acid changes at other sites). Unfortunately, in vivo recombination assays are the only way to get at this question.

Please let me reinstate my overall satisfaction with the paper. I really believe the authors provide here a novel approach to understand the ecology of cassettes that makes a lot of sense.

>>>> We appreciate the Reviewer's comment.

Reviewer #3 (Remarks to the Author):

The paper by Ghaly et al describes a study modelling the evolutionary sequence and structure conservation of integron attC sites across taxa and environments. The authors find that 1) taxonomic origins of integron gene cassettes can be predicted from the structure and sequence of their attC sites, and 2) attC sites cluster according to host environment rather than host phylogeny. They use these two findings to predict the origins for 28 antibiotic resistance cassettes spread by mobile integrons and predict the source environments of resistance cassettes. Both of these findings are new and reflect a novel approach to identify the origins of antibiotic resistance genes (cf. for example Ebmeyer et al in *Communications Biology*, which the authors cite). Since the origins for resistance genes are interesting both from a theoretical standpoint and for potential mitigation of future resistance threats, this is of great interest to the field.

The methods are sound and reproducible. I usually have criticisms about bioinformatic methods, but I could not find any critical flaws in the authors' approach. That said, I do wonder why the authors have chosen to make assignments using e-values (which is dependent on database size) rather than score or score per length? Is there any good reason for this? Note that using e-values is not scalable to big datasets.

>>>>> We thank the Reviewer for raising this valid point. We completely agree that using the bit score instead of e-value would be more appropriate. As such, we have re-done the analysis using scores instead of e-values. This has led to modifications in the predictions which have been incorporated in Table 1 and in the text (lines 237-254).

However, I do find the manuscript a bit hard to follow. I don't feel that there is always a logical flow between the different paragraphs. It is hard to be specific, but I struggled a bit to get the pieces together.

>>>>> We have made changes throughout the manuscript to improve its readability. We believe that these changes have allowed for a more logical flow.

I am also not entirely convinced by the method the authors use to connect environments to the origins of certain resistance gene classes. This analysis could be made more robust by also adding in, for example, metagenomic data, and by covering a larger range of environments. The categories "marine", "freshwater/soil" (which by the way is incredibly broad) and "Xanthomonadales-like" do not at all consider host-associated environments, for example. While I think that the reasoning around taxonomic origins is sound and robust, I feel that this approach lacks the support that the authors think they have to make claims about source environments.

>>>> Although we believe that the Reviewer makes a very good point here, we feel that the suggestions made might not be the most applicable in this context.

In particular, we foresee some potential issues using metagenomic-derived *attC*s to infer their environmental clustering. The main issues being: (a) by using metagenomic data, we would have a limited ability to distinguish between cassettes on mobile integrons and those on sedentary chromosomal integrons. This opens up the possibility of drastically skewing the results. This is particularly concerning given the massive distribution of class 1 integrons and the potential for them to exist at very high copy numbers in a given sample. Based on experience, chromosomal integrons on the other hand are much harder to detect in metagenomic data and require massive amounts of data to extract even a few *attC*s; and (b) by using metagenomic data we would lose valuable taxonomic information. The strength of using isolate genomes is that we can differentiate between clustering based on host phylogeny and those based on similar environment. In a metagenomic sample, we would not be able to tell if the enrichment of one or a few taxa is influencing *attC* clustering more than the environment it was sampled from.

On the other hand, using isolate genomes instead of metagenomes has several benefits. These include: (a) being able to use Fig.2 to validate our CM predictions. We can see that *attC*s cluster tightly with the taxa that we had predicted with our CMs, further strengthening our CM predictions. This is something that we cannot achieve using metagenomic data; and (b) using isolate genomes is the same approach used in previous publications to infer the environmental clustering of integron integrases (e.g. Mazel. 2006. *Nat. Rev. Microbiol.*; Boucher et al. 2007. *Trends Microbiol.*; Cambray et al. 2011. *Mobile DNA*). Thus, by using consistent methods, it allows us to compare *IntI*s and *attC*s to gain insights into their evolutionary dynamics

As the Reviewer has pointed out, the *attC*s are grouped into a small number of broad-scale environmental groupings. However, we believe that this is necessary. Because a bacterial taxon can span across several specific environments, environmental groupings have to be kept quite broad. Also, they are the same groupings that are used for integron integrases in the previous publications listed above. Again, to draw evolutionary insights between *IntI*s and *attC*s, congruency is needed.

Regarding the robustness of our approach, our analysis shows that (a) *attC*s cluster together, and that this clustering is better explained by the predominant host environment than by the phylogenetic relatedness of the hosts, and (b) that particular *attC*s of resistance cassettes are structurally similar to SCI *attC*s from particular environments. Given that these resistance cassettes have almost certainly been captured from SCIs, we can infer the likely environment of those taxa based on their *attC* structure.

Aside from this, I only have a few minor comments:

- Line 28: Unsure what the authors are implying with “still” here. Is that surprising?

>>>> Here, we are stating that we found cassettes within SCIs of the very taxa that we had predicted using our CMs. It is surprising given the small chance of those isolates having actually been sequenced, given that different but closely related strains often have completely different cassette arrays. Though, we make this point more because it strongly supports our predictions.

- Line 31: Not necessarily sources, could also be the environment the cassette provides the most advantage in.

>>>> Environments that provide the most advantage is where we would expect to see these cassettes at the greatest abundance/frequency. This is not what we have shown (i.e.

we do not know if aminoglycoside and beta-lactam resistance cassettes are most abundant in marine environments). Rather, what we have observed in this study is that most of these cassettes have attCs that are structurally similar to chromosomal attCs of marine bacteria. Thus, we can infer that cassettes with these attCs have also originated in marine bacteria.

- Line 104-108: This is not clear until after reading the rest of the paper. Perhaps this could be rephrased?

>>>> We thank the Reviewer for pointing this out. As suggested, we have rephrased this section (lines 104-109).

- Line 206: Likely to what? This sentence seems incomplete.

>>>> We have modified this sentence to “likely to occur” (Line 207)

- Line 225: As noted later by the authors, this is a substantial under-prediction. I would like the authors to comment a bit further on the implications of this under-prediction.

>>>> As suggested by the Reviewer, we have discussed the implications of this (lines 226-229).

- Line 264: Very minor comment which is not that important for the paper, but how do the authors define emergence? All of these resistance genes seem to have been pre-existing, so what is the emergence event, as defined by the authors of this paper?

>>>> Here, we are speaking more generally in which a resistance gene has ‘emerged’ in human settings and continue to spread and become problematic.

- Line 329-330: I don’t think that the authors do any effort to test whether taxa that are not closely related by descent have more similar intI genes and attC structures, and I think they should. A formal test of that would strengthen the conclusions of the paper.

>>>> By “more similar” we are signifying that they cluster close together in either the structure-based attC tree or the IntI phylogeny. For example, some taxa that are in separate phyla can be seen to cluster together (e.g. Cyanobacteria and Deltaproteobacteria), while taxa that are within the same phylum can be found on divergent clades (e.g. Pseudomonadales and Vibrionales).

- Line 353-354: Can the knowledge the authors generate really allow us to predict hotspots for the emergence of the next generation of resistance genes, if the origin is as wide as “the marine environment”? Can the authors connect this to any credible intervention? How does this knowledge make us more likely to be able to mitigate future resistance development?

>>>> Here, we present the first steps at predicting the sources of cassettes. The next step would be investigating these predictions further. Particularly for the ‘Xanthomonadales-like’ clade and the possibility that plant-associated environments are significant sources. Plants are often consumed raw and provide a direct route for cassettes to be picked up by class 1 integrons, which are ubiquitous in human guts within developed countries. It might also be a good place to screen for resistance to new antibiotics that are currently being developed (i.e. via a functional metagenomic approach).

Reviewers' Comments:

Reviewer #1:

Remarks to the Author:

The authors have responded to all the referee requests in a satisfactory manner. I have a slight different point of view with the authors concerning the co-evolution between integrases and attC sites. I agree with the authors that it would be interesting to confirm this hypothesis by performing in vivo recombination experiments using several integrases and attC sites. Nevertheless, I am perfectly aware that performing this type of experiment is beyond the scope of the current study. In conclusion, I think that the paper can be published in *Com Biology*.

Reviewer #2:

Remarks to the Author:

I am overall satisfied with the rebuttal. Authors have answered my questions and agreed to produce figures and tables I asked for. Nevertheless, the supplementary figure with labeled attC sites is not fully satisfactory because the label does not allow to identify univocally the cassette. Can the authors change these labels for the cassette names in Partridge's list?

With this correction I think the manuscript is ok.

Reviewer #3:

Remarks to the Author:

I am very happy with the changes the authors have made to the manuscript, and with the thorough explanations they have offered when they have decided to not to implement changes in response to my comments. I think that this very interesting manuscript is basically ready to be published. I have one minor issue and one final note on the revision:

I still think that the statement on line 347 in the revised manuscript that "taxa that are not necessarily closely related by descent, have more similar intI genes and attC structures" would benefit from a proper statistical test. Not that this really precludes the publication of the paper, but it would be elegant.

My final note is that I am still not that convinced that mitigations based on this work are feasible without much more fine-grained data on where these cassettes come from. But perhaps that does not need to be said in the paper.

Great work from the authors, both with the initial study and the revision!

Response to Reviewers

Reviewer #1 (Remarks to the Author):

The authors have responded to all the referee requests in a satisfactory manner. I have a slight different point of view with the authors concerning the co-evolution between integrases and attC sites. I agree with the authors that it would be interesting to confirm this hypothesis by performing *in vivo* recombination experiments using several integrases and attC sites. Nevertheless, I am perfectly aware that performing this type of experiment is beyond the scope of the current study. In conclusion, I think that the paper can be published in *Com Biology*.

Reviewer #2 (Remarks to the Author):

I am overall satisfied with the rebuttal. Authors have answered my questions and agreed to produce figures and tables I asked for. Nevertheless, the supplementary figure with labeled attC sites is not fully satisfactory because the label does not allow to identify univocally the cassette. Can the authors change these labels for the cassette names in Partridge's list?

With this correction I think the manuscript is ok.

Response: We have now adjusted the labels to have the same gene names from Partridge's list and also included the accessions from Partridge's list

Reviewer #3 (Remarks to the Author):

I am very happy with the changes the authors have made to the manuscript, and with the thorough explanations they have offered when they have decided to not to implement changes in response to my comments. I think that this very interesting manuscript is basically ready to be published. I have one minor issue and one final note on the revision:

I still think that the statement on line 347 in the revised manuscript that "taxa that are not necessarily closely related by descent, have more similar intI genes and attC structures" would benefit from a proper statistical test. Not that this really precludes the publication of the paper, but it would be elegant.

Response: We understand Reviewer 3's point, but feel that no statistical test is required on the basis that, for example, some taxa that are in separate phyla can be seen to cluster together (e.g. Cyanobacteria and Deltaproteobacteria), while taxa that are within the same phylum can be found on divergent clades (e.g. Pseudomonadales and Vibrionales). The point being that taxa that cluster together are not necessarily closely related.

My final note is that I am still not that convinced that mitigations based on this work are feasible without much more fine-grained data on where these cassettes come from. But perhaps that does not need to be said in the paper.

Response: Here, we agree with Reviewer 3. Perhaps future studies can build on our findings here to generate more fine-grained data on where the cassettes are coming from.

Great work from the authors, both with the initial study and the revision!